# EARLY FLOWERING 3 interactions with PHYTOCHROME B and PHOTOPERIOD1 are critical for the photoperiodic regulation of wheat heading time

**Maria Alejandra Alvarez**[1,2], **Chengxia Li**[1,2], **Huiqiong Lin**[1,2], **Anna Joe**[1,2], **Mariana Padilla**[1], **Daniel P. Woods**[1,2], **Jorge Dubcovsky**[1,2]*

1 Department of Plant Sciences, University of California, Davis, California, United States of America,
2 Howard Hughes Medical Institute, Chevy Chase, Maryland, United States of America

* jdubcovsky@ucdavis.edu

**Data Availability Statement:** Mutant lines were deposited in the National Small Grains Collection

## Abstract

The photoperiodic response is critical for plants to adjust their reproductive phase to the most favorable season. Wheat heads earlier under long days (LD) than under short days (SD) and this difference is mainly regulated by the *PHOTOPERIOD1* (*PPD1*) gene. Tetraploid wheat plants carrying the *Ppd-A1a* allele with a large deletion in the promoter head earlier under SD than plants carrying the wildtype *Ppd-A1b* allele with an intact promoter. Phytochromes *PHYB* and *PHYC* are necessary for the light activation of *PPD1*, and mutations in either of these genes result in the downregulation of *PPD1* and very late heading time. We show here that both effects are reverted when the *phyB* mutant is combined with loss-of-function mutations in *EARLY FLOWERING 3* (*ELF3*), a component of the Evening Complex (EC) in the circadian clock. We also show that the wheat ELF3 protein interacts with PHYB and PHYC, is rapidly modified by light, and binds to the *PPD1* promoter *in planta* (likely as part of the EC). Deletion of the ELF3 binding region in the *Ppd-A1a* promoter results in *PPD1* upregulation at dawn, similar to *PPD1* alleles with intact promoters in the *elf3* mutant background. The upregulation of *PPD1* is correlated with the upregulation of the florigen gene *FLOWERING LOCUS T1* (*FT1*) and early heading time. Loss-of-function mutations in *PPD1* result in the downregulation of *FT1* and delayed heading, even when combined with the *elf3* mutation. Taken together, these results indicate that ELF3 operates downstream of *PHYB* as a direct transcriptional repressor of *PPD1*, and that this repression is relaxed both by light and by the deletion of the ELF3 binding region in the *Ppd-A1a* promoter. In summary, the regulation of the light mediated activation of *PPD1* by ELF3 is critical for the photoperiodic regulation of wheat heading time.

## Author summary

The coordination of reproductive development with the optimal season for seed production is critical to maximize grain yield in crop species. Plants can perceive the length of

under ID numbers PI 701905 (Kronos-PS, introgression of photoperiod sensitive allele *Ppd-A1b*), PI 701906 (Kronos *elf3 phyB* combined knock-outs), and PI 701907 (Kronos elf3 ppd1). Additional information about these accessions and/ or seed requests can be done at GRIN-Global https://npgsweb.ars-grin.gov/gringlobal/search. All other relevant data are within the paper and its Supporting Information files.

**Funding:** J.D. acknowledges support from the Howard Hughes Medical Institute (https://www. hhmi.org/) and by competitive Grants 2016-67013-24617 and 2022-68013-36439 (WheatCAP) from the United States Department of Agriculture, National Institute of Food and Agriculture (https:// nifa.usda.gov/). D.P.W. was a Howard Hughes Medical Institute Post-Doctoral Fellow of the Life Sciences Research Foundation. The funders played no role in this research beyond providing the funding.

**Competing interests:** The authors have declared that no competing interests exist.

the day or night (photoperiod) and use this information to anticipate seasonal changes. In most eudicot plants, *CONSTANS* plays a central role in the perception of photoperiod, but in wheat the main photoperiod gene is *PHOTOPERIOD1* (*PPD1*). In this study, we show that the clock gene *EARLY FLOWERING 3* (*ELF3*) regulates the phytochrome-mediated light activation of *PPD1*. Loss-of-function mutations in *ELF3* result in the upregulation of *PPD1* at dawn, and in early heading under both long and short days, even in the absence of *PHYB*. A deletion in the *PPD1* promoter including an ELF3 binding region also results in earlier heading under short days, indicating that ELF3 acts as a direct transcriptional repressor of *PPD1*. This study shows that *ELF3* plays a critical role in the wheat photoperiod pathway by regulating the light signal between the phytochromes and *PPD1*. ELF3 provides an additional entry point to engineer heading time in wheat, an important trait for the development of better adapted varieties to a changing environment.

## Introduction

Crop productivity depends on the precise alignment of flowering with the most favorable season for seed production and grain filling. For that to happen, plants need to anticipate the proximity of the favorable season to initiate the reproductive phase in a timely manner. The duration of days and nights (photoperiod) provides seasonal information that plants have evolved to sense and use to this end [1].

In Arabidopsis and other eudicot species *CONSTANS* (*CO*) is the main photoperiodic gene [2–4], but recent studies in wheat have shown that combined loss-of-function mutations in *CO1* and its close paralog *CO2* (*co1 co2*) have a limited effect on wheat photoperiodic response [5]. The earlier heading of wheat under long days (LD) than under short days (SD) is mainly regulated by the *PHOTOPERIOD1* (*PPD1*) gene, also known as *PSEUDO RESPONSE REGULATOR 37* (*PRR37*) [6,7]. Large deletions in the promoter of the A-genome (*Ppd-A1a*) [8] or the D-genome (*Ppd-D1a*) [6] homoeologs result in altered gene expression and earlier heading under SD than in genotypes carrying the wildtype alleles (*Ppd1b*). These deletions share a common ~900 bp region that has been hypothesized to include binding sites for one or more transcriptional repressors [8]. Wheat genotypes carrying the *Ppd1b* alleles head extremely late under SD and are referred to as photoperiod sensitive (PS). Genotypes carrying any dominant *Ppd1a* allele are referred to as photoperiod insensitive (PI), even though they still head significantly earlier under LD than under SD. Differences between PS and PI wheats have been shown to be associated with grain productivity in different environments [9,10].

The acceleration of heading time by *PPD1* requires its transcriptional activation by light, which is mediated by phytochromes PHYB and PHYC [11–13]. Phytochromes are dimeric proteins that function both as red and far-red light sensors and as temperature sensors [14,15]. Phytochromes can transition from an inactive "Pr" form to an active "Pfr" form upon absorption of red light. In the darkness or upon absorption of far-red light, phytochromes revert to the inactive "Pr" state [16]. The rate of this dark reversion is temperature-dependent providing phytochromes the ability to perceive both light and temperature signals and integrate them to regulate photo- and thermo-morphogenetic processes [15].

The role of wheat *PHYB* and *PHYC* in the induction of *PPD1* is particularly evident in night-break (NB) experiments, where a 15-min pulse of white light in the middle of the SD long nights (16h) is sufficient to induce *PPD1* expression. However, at least two weeks of NBs (or LDs) are necessary to accelerate heading time [17]. Although 15 NBs accelerate heading time of PS wheat plants by more than three months, the same NBs in wheat lines carrying

*phyB* or *phyC* loss-of-function mutations fail to induce *PPD1* expression and plants head extremely late (~160–170 d) [17]. Similarly, wheat lines carrying *ppd1* loss-of-function mutations are late heading under both LD (~100–130 d) [5,18] and NB (>150 d) conditions [17]. Taken together, these results indicate that *PPD1* is central to the wheat NB and photoperiodic responses and that the duration of the night is critical for the perception of photoperiodic differences in this species.

The transcriptional activation of *PPD1* by light results in the induction of *FLOWERING LOCUS T1* (*FT1*) expression in the leaves [5,17]. *FT1* encodes a protein (florigen) that is thought to migrate through the phloem to the shoot apical meristem in wheat as shown in Arabidopsis and rice [19,20]. FT1 interacts with 14-3-3 and FDL proteins to form a florigen activation complex that binds to the promoter of the MADS-box transcription factor *VERNALIZATION1* (*VRN1*), which is critical to trigger the transition of the shoot apical meristem from the vegetative to the reproductive phase [21,22]. The upregulation of *VRN1* results in the downregulation of the flowering repressor *VERNALIZATION2* (*VRN2*), generating a positive feedback loop that accelerates heading time [23–25]

Significant genetic interactions have been detected between *PPD1* and *EARLY FLOWERING 3* (*ELF3*) in the regulation of heading time in both wheat and barley [26–28]. In Arabidopsis, it has been shown that ELF3 is part of the Evening Complex (EC), an important component of the circadian clock, that also includes LUX ARRHYTHMO (LUX) and EARLY FLOWERING 4 (ELF4) [29]. ELF3 functions as an adaptor between LUX and ELF4 and as a hub for multiple protein interactions [29]. In barley and wheat, loss-of-function mutations in *ELF3* [26,27,30] or *LUX* [31–33] result in very early heading under both LD and SD conditions, suggesting that in these species the EC is critical for the photoperiodic response. Natural allelic variation at *ELF3* has also been associated with differences in heading time in diploid *Triticum monococcum* and in hexaploid wheat [26,34,35].

In Arabidopsis, the expression of *ELF3*, *ELF4*, and *LUX* overlaps and peaks at dusk, maximizing their repressive effect early in the night [36]. Chromatin immunoprecipitation studies have shown that the EC acts as a direct transcriptional repressor of the circadian morning loop genes *PSEUDORESPONSE REGULATOR 7* (*PRR7*) and *PRR9* in Arabidopsis [37,38] and *PRR37*, *PRR73*, *PRR95*, *GI*, and *GHD7* in rice [39] (*GHD7* is the rice ortholog of the wheat LD flowering repressor *VRN2* [40,41]). Since Arabidopsis *PRR7/PRR3* are the closest homologs of wheat and barley *PPD1*, and *elf3* mutants in barley and *Brachypodium* show altered *PPD1* expression profiles [27,42], we hypothesized that a similar regulation of *PPD1* by ELF3 may contribute to their observed epistatic interactions on heading time in the temperate cereals.

In this study, we show that the wheat ELF3 protein interacts with PHYB and PHYC, is modified by light, and binds to the promoter of *PPD1*. We also show that the repressed expression of *PPD1* in the *phyB* mutant is restored in the *elf3 phyB* combined mutant, which flowers almost as early as the *elf3* mutant. By contrast, the combined *elf3 ppd1* mutant heads significantly later than *elf3*. Based on our results, we propose that ELF3 acts as a direct repressor of *PPD1* and plays a critical role in the regulation of the light signals between the phytochromes and *PPD1*.

## Results

### The heading time delay of the *phyB* mutant mostly disappears in the absence of *ELF3*

Given the documented interactions between ELF3 and PHYB in *Arabidopsis* [29,43,44], we hypothesized that the large delay on wheat heading time observed in the *phyB* mutant in previous studies [12,13] could be mediated by *ELF3*. To test this, we generated loss-of-function

mutants *phyB*, *elf3* and *elf3 phyB* (Fig A in S1 Text) in two different tetraploid wheat cultivar Kronos backgrounds, one carrying the photoperiod insensitive (PI) *Ppd-A1a* allele with a promoter deletion (Fig A in S1 Text) and the other one carrying the photoperiod sensitive (PS) *Ppd-A1b* allele with an intact promoter. We evaluated these lines in growth chambers for heading time under SD (8 h light, 16 h dark) and LD (16 h light, 8 h dark).

Under LD, strong genetic interactions between *ELF3* and *PHYB* were detected in both PS and PI plants (Fig 1A and 1B). The *phyB* mutant failed to head before the experiment was terminated at 160 d, but the combined *elf3 phyB* mutant headed in 43 d in PI (Fig 1A) and in 49 d in PS (Fig 1B). In both genetic backgrounds, the *elf3* mutation accelerated heading time by 11 d in the presence of the wildtype *PhyB* allele, but more than 110 days in the *phyB* mutant background. The early heading time of the *elf3* mutant was associated with a reduced number of spikelets per spike as reported previously [26], but this number was restored to wildtype levels in *elf3 phyB* (Fig B in S1 Text and Data H in S1 Data).

Under SD, PI plants carrying the *Elf3* and *PhyB* wildtype alleles headed in 97 d (Fig 1C), whereas PS plants did not head (Fig 1D), as in previously published studies [5]. Similarly, both PI and PS *phyB* mutants failed to head before the end of the experiment. By contrast, plants carrying only the *elf3* mutant allele headed very early in both PS and PI backgrounds (~46 d). Interestingly, when the *elf3* mutation was present, the *phyB* loss-of-function mutation was associated with a non-significant 2 d delay in heading time in PI (Fig 1C), but with a 4.5 d significant acceleration in PS (Fig 1D) relative to the wildtype *PhyB* allele.

The same data used in Fig 1A–1D is reorganized in Fig 1E to facilitate comparisons among photoperiods and *PPD1* alleles in the different mutant backgrounds. The wildtype plants headed significantly later under SD than under LD, but these differences were greatly reduced in the absence of *ELF3*. In the *elf3* single mutant, SD was associated with a significant delay in heading time in both PI (6.3 d) and PS (3.9 d). However, in the combined *elf3 phyB* mutant, SD was associated with a significant delay in heading time in PI (5.9 d), but a significant acceleration in PS (7.8 d, $P < 0.001$, Fig 1E). Earlier heading under SD than under LD has been reported before both for the *phyB* and *phyC* mutants in Kronos [11,12], an interesting result considering that wheat is a LD-plant.

In summary, the heading time results indicate that *i*) the large delays in heading time caused by the *phyB* mutation are mediated in large part by *ELF3*, *ii*) the *elf3* mutant is still able to respond to photoperiod, and *iii*) the *elf3 phyB* mutant heads earlier under LD than under SD in the PI background, but later in the PS background.

## *PPD1* and *FT1* are downregulated in *phyB* but expression is restored in *elf3 phyB*

To better understand the strong genetic interaction observed between *ELF3* and *PHYB* on heading time, we explored the expression of *PPD-A1*, *PPD-B1*, and *FT1* in *phyB* and *elf3 phyB* mutants grown under LD (Fig 2). In PI, the *PPD-A1a* expression peak observed in the wildtype at dawn was no longer detected in the *phyB* mutant (Fig 2A), which also showed a significant downregulation of *PPD-B1* from ZT8 to ZT20 (Fig 2B). The downregulation of both *PPD1* homoeologs in *phyB* was reflected in reduced transcript levels of *FT1* (Fig 2C) and *VRN1* (Fig C in S1 Text) which, together with the upregulation of the *VRN2* repressor (Fig C in S1 Text), resulted in late heading (Fig 1). The PI *phyB* mutant also showed upregulation of *CO1* (ZT0-ZT8) and *CO2* (ZT4-ZT8) (Fig C in S1 Text). In PS, the *phyB* mutant showed an even more pronounced downregulation of *PPD-A1b* and *PPD-B1*, which resulted in undetectable *FT1* levels (Fig 2D–2F).

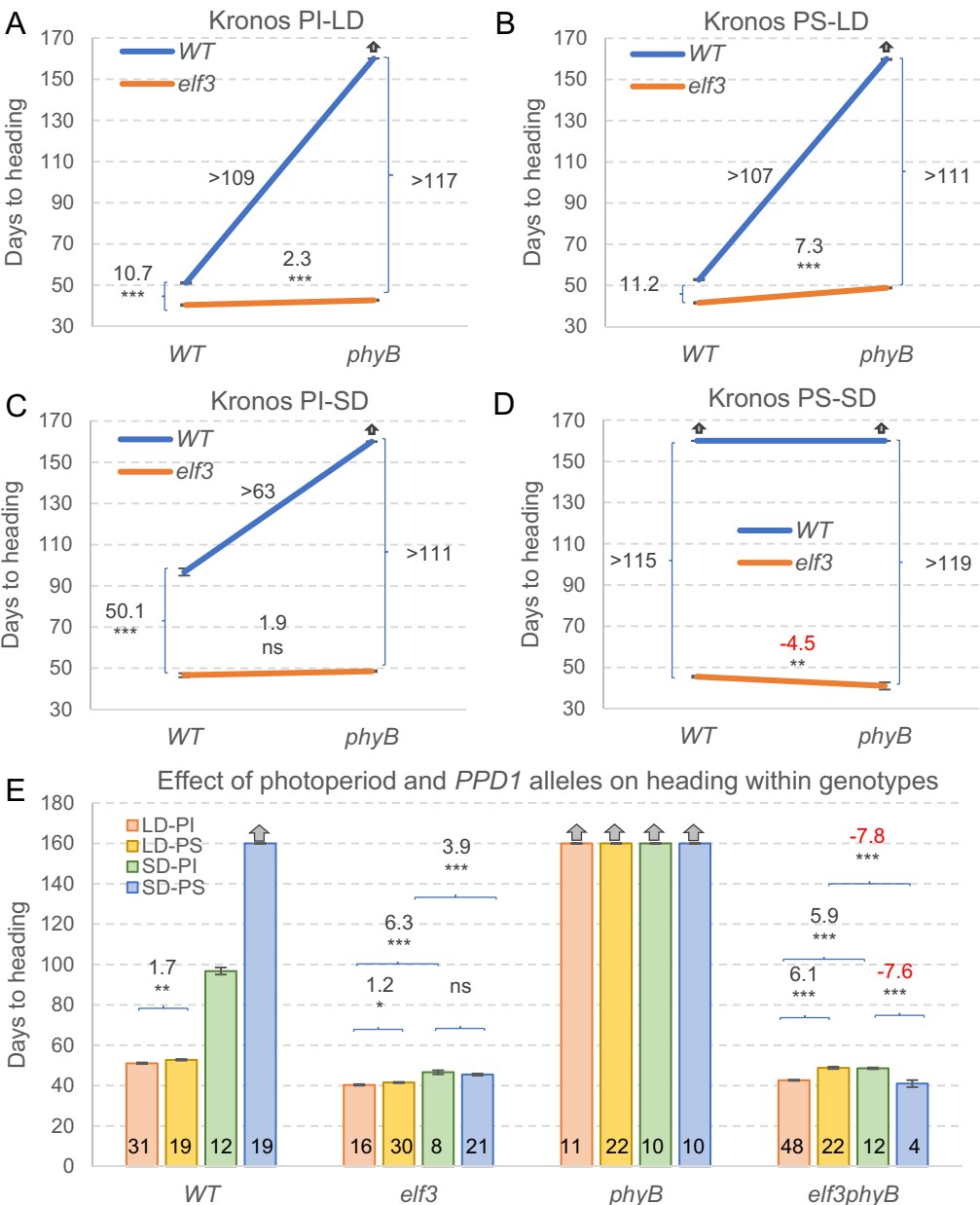

**Fig 1. Effect of *phyB* and *elf3* mutations on wheat heading time.** (**A-B**) Long days (LD). (**C-D**) Short days (SD). (**A and C**) Kronos photoperiod insensitive (PI) background and (**B and D**) photoperiod sensitive (PS) background. (**E**) Effect of the different photoperiods and genetic backgrounds on heading time within each mutant combination. Error bars are s.e.m. Numbers in the base of the bars indicate the number of plants analyzed in each genotype / photoperiod combination. * = *P* < 0.05, ** = *P* < 0.01, *** = *P* < 0.001 (Tukey tests). No statistical tests were performed for the plants that failed to head at 160 d when the experiment was terminated (indicated by gray arrows). Raw data and statistics are in Data A in S1 Data.

The altered expression profile of *PPD-A1a* also resulted in an altered *PPD-B1* expression profile (Fig 2B and 2E), a phenomenon that has been previously reported for different *PPD1a* alleles in hexaploid wheat [45,46]. These results suggest the existence of a feed-back regulatory loop where some of the genes differentially regulated by the *PPD1a* alleles affect the regulation of the other *PPD1b* homoeologs.

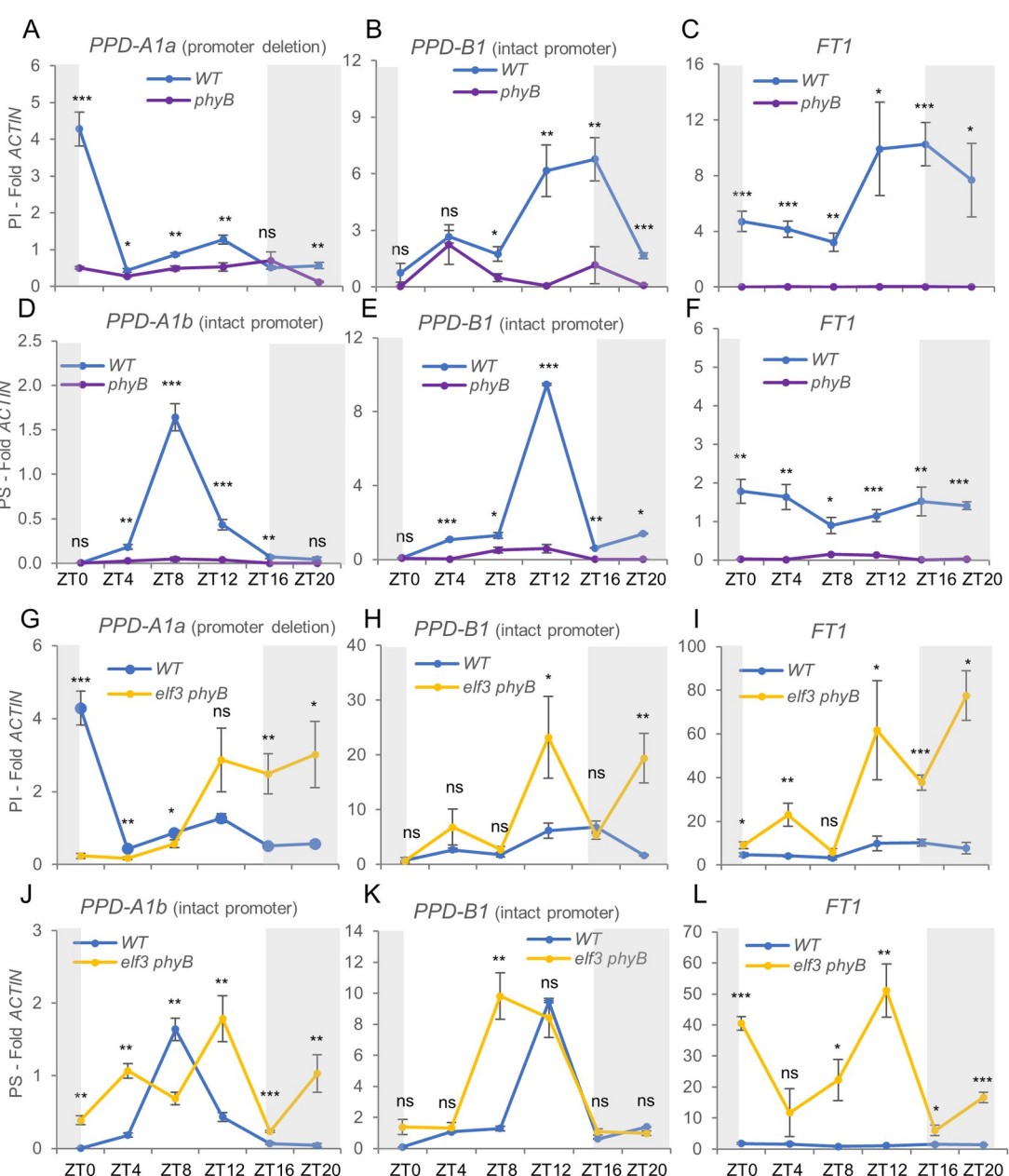

**Fig 2. Transcript levels of *PPD1* and *FT1* in *phyB* and *phyB elf3* mutants.** Samples were collected from leaves of 5-week-old Kronos photoperiod insensitive (PI) and sensitive (PS) plants grown under LD. (**A-C**) Wildtype *vs. phyB* in PI. (**D-E**) Wildtype *vs. phyB* in PS. (**G-I**) Wildtype *vs. elf3 phyB* in PI. (**J-L**) Wildtype *vs. elf3 phyB* in PS. (**A & G**) *PPD-A1a* allele with a deletion in the promoter. (**D & J**) *PPD-A1b* allele with intact promoter. (**B, E, H, & K**) *PPD-B1* photoperiod sensitive allele. (**C, F, I, & L**) *FT1* (both homoeologs). Expression in single *elf3* mutants is discussed later in the *ELF3 x PPD1* section. *ACTIN* was used as endogenous control. Error bars are s.e.m based on 5 biological replications. ns = not significant, * = $P < 0.05$, ** = $P < 0.01$, *** = $P < 0.001$ based on t-tests between mutants and wildtype at the different time points. Raw data and statistics are available in Data B in S1 Data.

The transcriptional repression of *PPD1* in the *phyB* mutant was greatly reduced in the *elf3 phyB* combined mutant (Fig 2G, 2H, 2J and 2K), resulting in higher *FT1* (Fig 2I and 2L) and *VRN1* (Fig C in S1 Text) transcript levels than in *phyB*. In *elf3 phyB*, *VRN2* transcript levels remained elevated, whereas the upregulation observed in *phyB* for *CO1* and *CO2* at ZT4-ZT8 was no longer

significant. In addition, *CO1* showed a significant downregulation relative to the wildtype at dawn in *elf3 phyB* (Fig C in S1 Text). In summary, the increased expression of *PPD1*, *FT1* and *VRN1* in *phyB elf3* correlates well with its early heading relative to *phyB* and the wildtype (Fig 1).

## ELF3 interacts with PHYB and PHYC in yeast-two-hybrid assays

*ELF3* is expressed at high levels in the leaves (4 to 15-fold *ACTIN*) and shows similar transcriptional profiles under SD and LD (Fig D in S1 Text). The *phyC* mutant showed no significant differences with the wildtype on *ELF3* expression at any time point, while the *phyB* mutant showed a marginally significant downregulation ($P < 0.05$) between ZT12 and ZT20 relative to the wildtype (Fig D in S1 Text and Data J in S1 Data).

Since the transcriptional regulation of *ELF3* by *PHYB* and *PHYC* was insufficient to explain the observed genetic interactions in heading time, we looked at possible interactions at the protein level using yeast-two-hybrid (Y2H) assays. The full-length coding region of the *ELF3* gene was fused to the GAL4 DNA-binding domain and used as bait. The phytochrome genes *PHYC* and *PHYB* were each divided into two parts encoding the N-terminal photosensory and C-terminal regulatory regions, respectively. The N-terminal region included 625 amino acids in PHYB (N-PHYB) and 600 amino acids in PHYC (N-PHYC). The C-terminal region encoded amino acids 626–1166 in PHYB (C-PHYB) and amino acids 601 to 1139 in PHYC (C-PHYC).

ELF3 showed strong interactions with both C-PHYB and C-PHYC. However, its interaction with the N-PHYB was much stronger than with N-PHYC (Fig 3). Autoactivation tests showed no interaction with empty vectors for ELF3 bait and PHYB preys (Fig 3). N-PHYC and C-PHYC preys have been tested previously and showed no autoactivation [11].

## The ELF3 protein is modified by light

Given that the ELF3 protein directly interacts with phytochromes, we hypothesized that light may affect its stability. To monitor the ELF3 protein, we generated transgenic Kronos PS lines constitutively expressing a C-terminal HA-tagged ELF3 under the maize *UBIQUITIN* promoter (UBI::ELF3-HA). The transgene was expressed at high levels in the leaves of both Kronos PS and the *elf3* mutant, and was associated with the downregulation of *PPD1* and *FT1* and delayed heading, including in plants carrying the *elf3* mutation (Fig E in S1 Text). These results confirmed that the UBI::ELF3-HA transgene is functional and can complement the *elf3* mutant phenotype. The ELF3-HA protein detected by immunoblotting using an anti-HA antibody was approximately 110 kDa, which is slightly higher than the predicted 88.5 kDa both in protein extracted from UBI::ELF3-HA transgenic plants and from transformed protoplasts (Fig F in S1 Text). This was the only band detected in the blots (except for rubisco) and it disappeared in the non-transgenic controls confirming that it is ELF3-HA. A higher than expected size for the ELF3 protein was also observed in western blots in rice [39].

We then looked at the stability of the ELF3 protein in UBI::ELF3-HA transgenic plants grown under LD (Fig 4A) and SD (Fig 4B) by immunoblotting using an anti-HA antibody. Under LD, two bands were detected during the day. The intensity of these bands varied during the day but were stronger later in the afternoon. The higher and more diffuse band disappeared in the dark, whereas the lower band became more intense after the lights were turned off (Fig 4A). Under SD, we also observed a strong lower ELF3 band 30 min after the lights were turned off, which faded away when the lights were turned on. A higher, fainter, and more diffused band fluctuated during the day (Fig 4B).

The difference between the two ELF3 bands is easier to see in the experiment in Fig 4C, where we grew plants under SD and, on the night of the experiment, we turned on the lights 2 h after the start of the night and kept them on during the subjective night (gray bar, Fig 4C). A

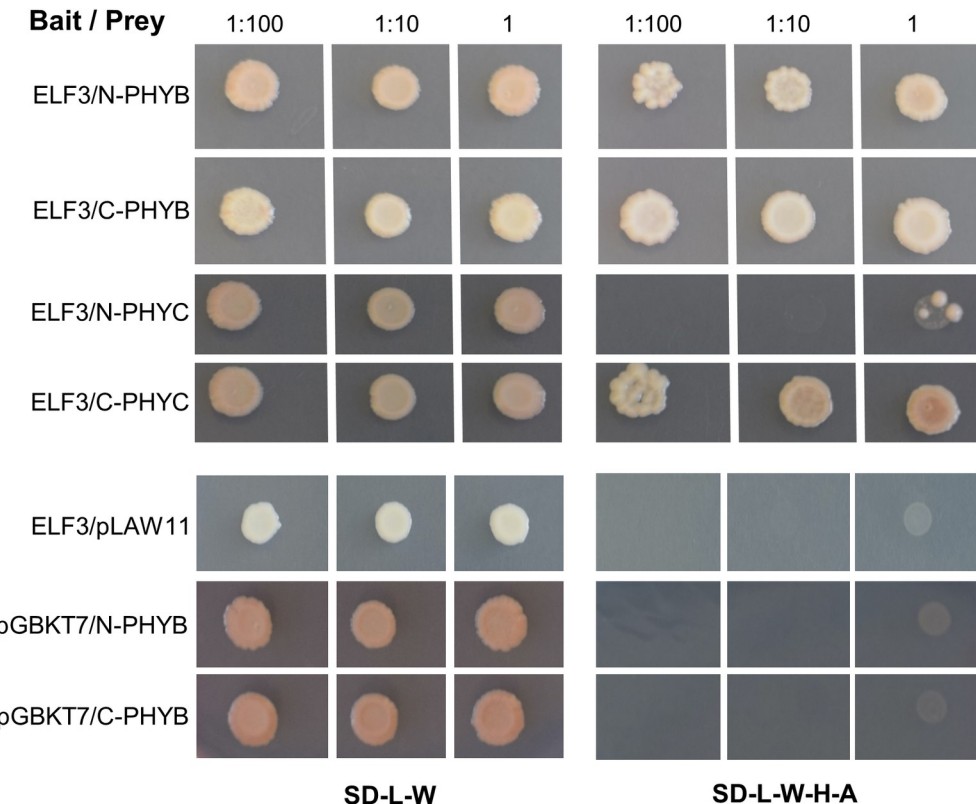

**Fig 3. Yeast-two-hybrid interactions between ELF3 and phytochromes PHYB and PHYC.** SD medium lacking Leucine and Tryptophan (-L-W) was used to select for yeast transformants containing both bait and prey vectors. Interactions were determined on SD media lacking Leucine, Tryptophan, Histidine and Adenine (-L-W-H-A). Autoactivation was tested for ELF3 bait using the empty prey vector pLAW11, and for N-PHYB and C-PHYB preys using the empty bait vector pGBKT7. We did not add a chromophore, so we are likely seeing the interaction with the inactive Pr form.

large increase in intensity of a higher and more diffused band and a decrease in intensity of the lower band were observed 10 min after the lights were turned on at ZT10 (Fig 4C), confirming that the transition between the lower and higher bands was mediated by light, rather than by the circadian clock alone. However, the intensity of the upper band 10 min after the lights were turned on was very different at ZT0 than at ZT10 (Fig 4C), suggesting that not only the light but also the time of the day is important in the regulation of the ELF3 protein. In the SD control where lights were not turned on during the night, the higher band was not detected in the dark (Fig 4D).

In RNA samples collected from leaves at the same time points as in Fig 4C and 4D, *PPD1* was significantly upregulated 30 min after the lights were turned on (Fig 4E, SD-int.), but remained low in the SD control where the night was not interrupted by light. These results suggest that the lower ELF3 band is the critical active form for the repression of *PPD1* during the night.

## Loss-of-function mutations in *PPD1* delay heading time in the absence of *ELF3*

The upregulation of *PPD1* in *elf3 phyB* relative to *phyB* (Fig 2) and its downregulation in the UBI::ELF3-HA transgenic plants (Fig E in S1 Text) suggested that ELF3 acts as a

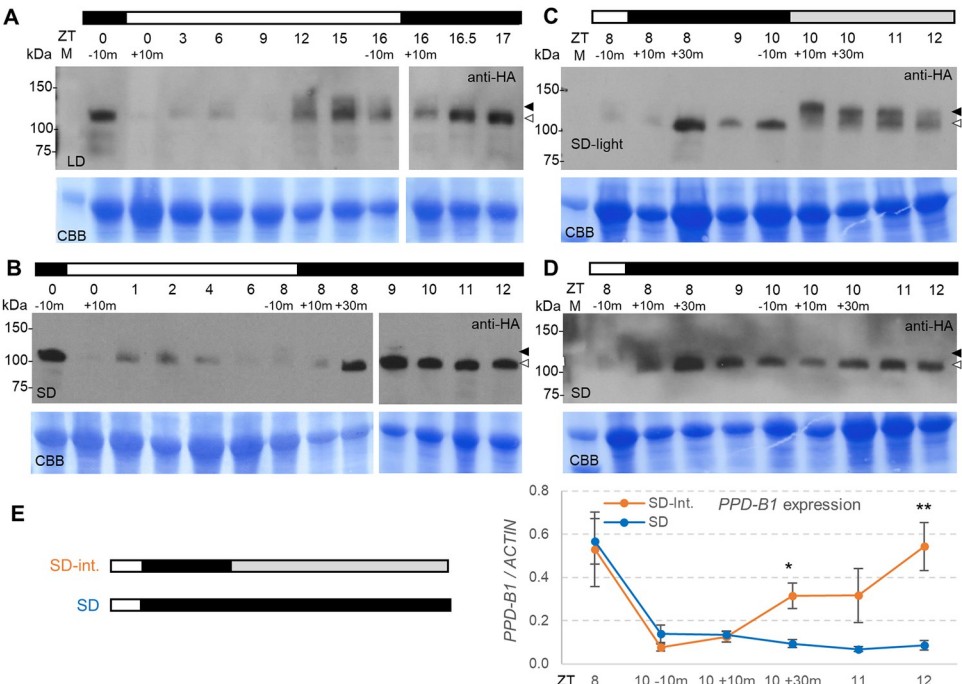

**Fig 4. ELF3-HA protein in UBI::ELF3-HA transgenic plants grown under LD, SD and SD interrupted night (SD-int.).** ELF3 protein levels were analyzed by immunoblotting using an anti-HA antibody (samples were harvested at the indicated ZT times). (**A**) Plants grown under LD. (**B**) Plants grown under SD. (**C**) Plants were grown under SD, but on the day when samples were collected, lights were turned on at ZT10, 2 h after the start of the night (SD-int.). (**D**) Plants under SD grown simultaneously with those in B but without turning on the light before sampling (collected at the same time points as in C). The black arrowhead indicates the higher and more diffuse band and the white arrowhead the sharper lower band detected in the dark. The bottom panel is a Coomassie Brilliant Blue (CBB) stained membrane used as a loading control. The white bar indicates lights on and the black bar lights off. The gray bar in C indicates that the lights were turned on during the subjective night. (**E**) Quantitative reverse transcription PCR (qRT-PCR) analysis of *PPD-B1* expression in leaves collected at the same time points as in C. Raw data and statistics are available in Data C in S1 Data.

transcriptional repressor of *PPD1*, so we combined loss of function mutations on both genes in the PI background to study their epistatic interactions on heading time. As controls, we included Kronos PI and the *elf3* mutant in the PI background.

For all genotypes, heading time was significantly later in SD than in LD, but the differences were reduced when the *elf3* mutant allele was present (Fig 5). Under LD, the *elf3* mutants headed 6 d earlier than the wildtype, while the *elf3 ppd1* combined mutants headed 20 d later than the wildtype, but still 36 d earlier than the *ppd1* lines. Under SD, the relative order of heading time of the genotypes was similar to the LD, but the differences were larger. The *elf3* mutant headed 36 d earlier than the wildtype, the *ppd1* lines failed to head by the time the experiment was terminated at 150 d, and *elf3 ppd1* combined mutants headed >48 d earlier than the single *ppd1* mutant (Fig 5).

A factorial ANOVA including *ELF3* (wildtype and *elf3*), *PPD1* (wildtype and *ppd1*) and photoperiod (SD and LD) revealed highly significant effects for these three factors and also for all two-way and three-way interactions ($P < 0.001$, Data D in S1 Data). These results indicate that *i)* there are strong genetic interactions between *ELF3*, *PPD1*, and photoperiod, *ii)* the *Ppd-A1a* allele can accelerate heading time in the absence of *ELF3* under both LD and SD, *iii)* *ELF3* can delay heading time in a *PPD1*-independent manner, and *iv)* there is a residual photoperiodic response that is independent of *PPD1* and *ELF3*.

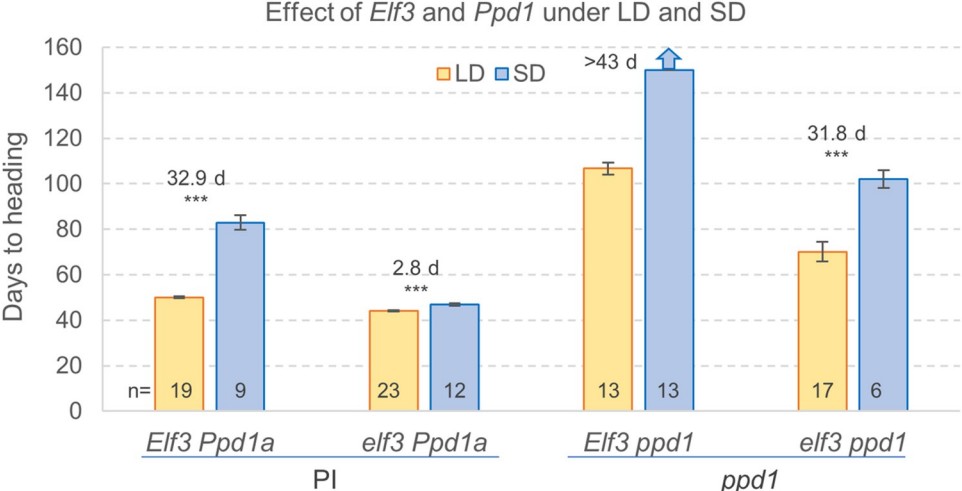

**Fig 5. Effect of *elf3* and *ppd1* mutations on heading time under LD and SD conditions.** Bars are means and error bars are s.e.m. The blue arrow on top of the *Elf3 ppd1* SD treatment indicates that plants did not head by the time the experiment was terminated at 150 d. Differences in heading times between SD and LD are indicated for each genotype above the bars with the corresponding *t*-tests (*** = *P* < 0.001). No *t*-test is provided for *Elf3 ppd1* because plants failed to head under SD. Numbers at the base of the bars indicate the number of plants analyzed in each genotype / photoperiod combination. Raw data and statistics are available in Data D in S1 Data.

## Effect of *elf3* mutations on *PPD1* and *FT1* expression

To understand the effect of the *elf3* mutations on heading time at the molecular level, we analyzed the transcript levels of *PPD1* and *FT1* in the youngest leaves of five-week-old Kronos PI and PS plants grown under LD (Fig 6). In the PI lines (Fig 6A–6C), the presence of two *PPD1* homoeologs with contrasting photoperiodic regulation provided interesting insights on the regulation of this gene. The *PPD-A1a* homoeolog showed a similar expression profile between the wildtype and the *elf3* mutant, with a strong peak at dawn (ZT0) (Fig 6A). By contrast, in the *PPD-B1* homoeolog this peak was present in the *elf3* mutant but disappeared in the wildtype. The *elf3* mutant also showed a significant upregulation of *PPD-B1* relative to wildtype at ZT20-ZT8 (Fig 6B), which was paralleled by a significant upregulation of *FT1* at ZT0-ZT4 (Fig 6C).

We hypothesized that the difference in the expression profiles between the two *PPD1* homoeologs at dawn was associated with the presence of the 1,027 bp deletion in the promoter of *Ppd-A1a* [8]. To test this hypothesis, we repeated the experiment in the PS lines, which carry the *Ppd-A1b* allele with an intact promoter. Indeed, we found that the ZT0 peak was absent in *PPD-A1b* (Fig 6D). Consistent with the earlier heading of PI relative to PS, the wildtype transcript levels of *FT1* were 2.6- to 8.5-fold higher in PI than in PS throughout the day (*P* = 0.028, Fig 6C and 6F). In the *elf3* mutant, *FT1* transcript levels were significantly higher than in the wildtype in both PI and PS, which agrees with their very early heading time (Fig 1).

## Effect of *ppd1* and *elf3* mutations on the expression of *VRN1*, *VRN2*, and *FT1*

To further characterize the interaction between *ELF3* and *PPD1* on heading time, we evaluated the expression levels of the main wheat flowering genes in five-week-old PI wildtype, *elf3*, *ppd1*, and *elf3 ppd1* plants grown under LD. The significant upregulation of *FT1* (Fig 7A–7C) at dawn in the *elf3* mutant was correlated with a significant upregulation of *VRN1* (Fig 7D–7F)

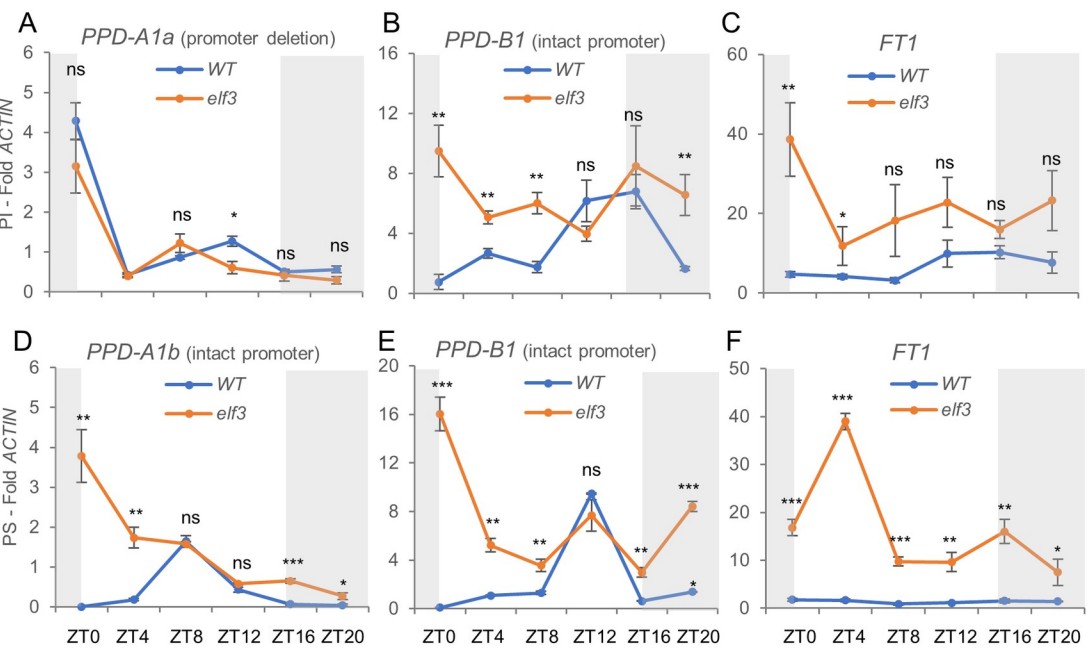

**Fig 6. qRT-PCR analysis of transcript levels of *PPD1* and *FT1* in wildtype and *elf3* mutants.** Leaf samples collected from 5-week-old Kronos photoperiod insensitive (PI) and sensitive (PS) plants grown under LD. (**A-C**) Wildtype *vs. elf3* in PI. (**D-F**) Wildtype *vs. elf3* in PS. (**A**) *PPD-A1a* allele with a deletion in the promoter associated with earlier heading under SD. (**D**) *PPD-A1b* allele with the intact promoter and late heading under SD. (**B & E**) *PPD-B1* with intact promoter in PI and PS. (**C & F**) *FT1* (both homoeologs combined). *ACTIN* was used as endogenous control. Error bars are s.e.m based on 5 biological replications. ns = not significant, * = P < 0.05, ** = P < 0.01, *** = P < 0.001 based on *t*-tests between mutants and wildtype at the different time points. Raw data and statistics are available in Data E in S1 Data.

at the same time point. Consistent with previous reports [28,39,42], the lack of a functional *ELF3* also resulted in the upregulation of the flowering repressor *VRN2* (Fig 7G–7I).

In the absence of functional copies of *PPD1*, the transcript levels of *FT1* and *VRN1* were almost undetectable, whereas *VRN2* was highly upregulated (Fig 7B, 7E and 7H). The transcription profiles of these three genes are consistent with the delayed heading time of the *ppd1* mutant (Fig 5). When the *ppd1* mutant was combined with *elf3*, the *FT1* transcripts remained severely downregulated (Fig 7C), but *VRN1* showed a gradual upregulation from ZT12 to ZT20 (Fig 7F), which is consistent with the earlier heading of *elf3 ppd1* relative to *ppd1* (Fig 5). The *VRN2* flowering repressor was upregulated to higher levels in *elf3 ppd1* than in the *elf3* or *ppd1* single mutants (Fig 7I). In summary, the upregulation of *VRN1* likely contributes to the accelerated heading time of the *elf3 ppd1* relative to *ppd1* (Fig 5) although this contribution may be partially offset by the increase in *VRN2* transcript levels.

Finally, we showed that in *elf3*, *CO1* transcript levels were significantly downregulated relative to the wildtype (Fig 7J), whereas those of *CO2* were not affected. *CO1* was upregulated in *ppd1* at all time points, and that upregulation was higher in the *elf3 ppd1* combined mutant (Fig 7K and 7L). The effect of *ppd1* on *CO2* was significant only at ZT4, whereas that of *elf3 ppd1* only at ZT12-ZT20.

It is interesting to point out that at dawn (ZT0), *PPD1*, *VRN1*, *VRN2*, *CO1*, and *CO2* are expressed at lower levels in *elf3 phyB* (Figs 2 and C in S1 Text) than in *elf3* (Figs 6 and 7), suggesting that at this time point *PHYB* is required for the upregulation of these genes in the *elf3* mutant background. However, given the complex feedback regulatory loops that exist among these genes, we do not know which of these effects are direct or indirect.

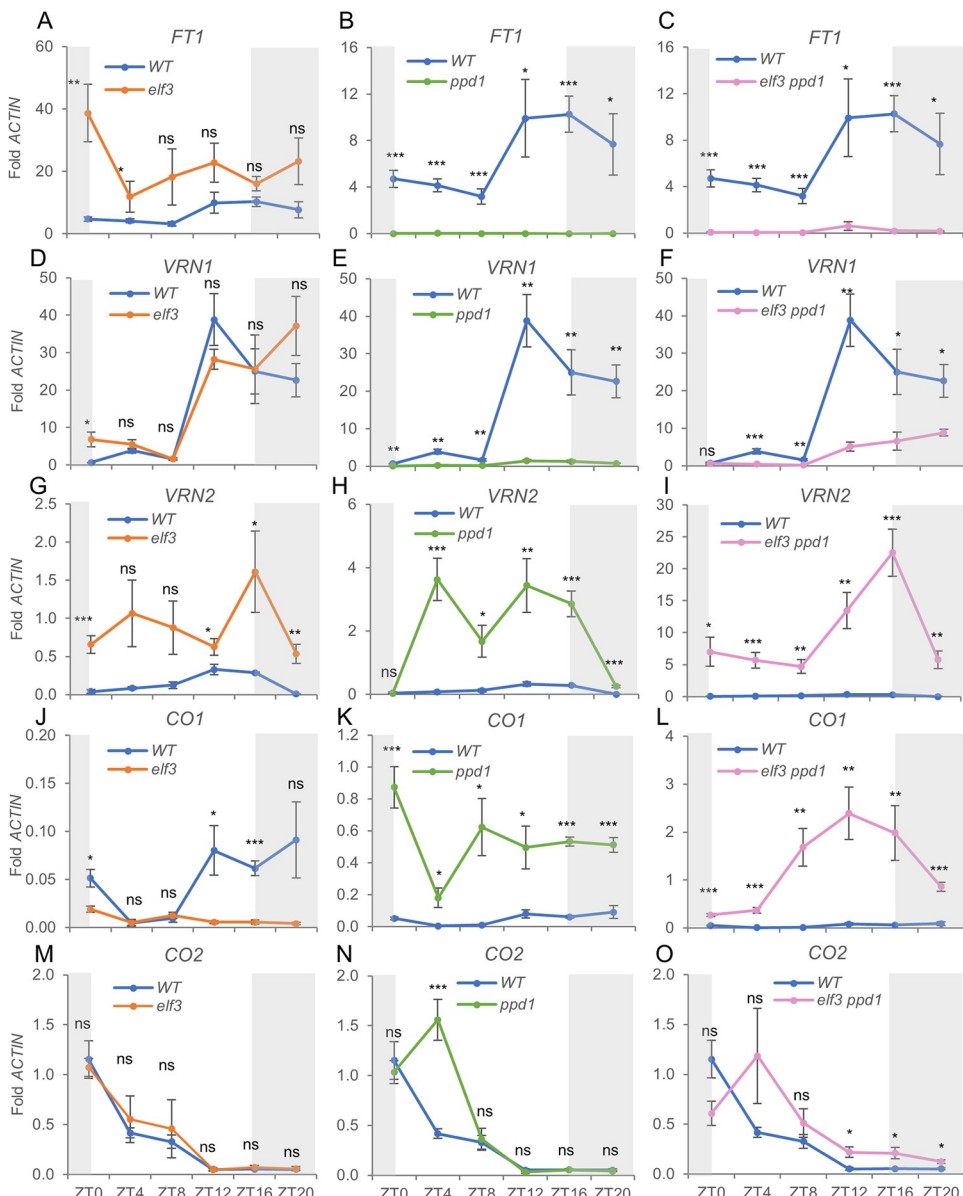

**Fig 7. Transcript levels of flowering genes *FT1*, *VRN1*, *VRN2*, *CO1*, and *CO2* in Kronos PI, *elf3*, *ppd1* and *elf3 ppd1* mutants.** (**A-C**) Flowering promoter gene *FT1*. (**D-F**) Flowering promoter gene *VRN1*. (**G-I**) Flowering repressor gene *VRN2*. (**J-L**) *CO1*. (**M-O**) *CO2*. (**A, D, G, J**, & **M**) Wildtype *vs. elf3*. (**B, E, H, K**, & **N**) Wildtype *vs. ppd1*. (**C, F, I, L**, & **O**) Wildtype *vs. elf3 ppd1*. Primers used for all genes amplify both homoeologs. The WT data is the same within each row but at different scales. Fig 7A is the same as Fig 6C, and is included again to facilitate comparisons. Error bars are s.e.m based on 5 biological replications. ns = not significant, * = $P < 0.05$, ** = $P < 0.01$, *** = $P < 0.001$ based on *t*-tests between mutants and wildtype at the different time points. Raw data and statistics are available in Data F in S1 Data.

## ELF3 binds directly to the *PPD1* promoter *in vivo*

The elevated expression levels of *PPD1* in the *elf3* mutant suggest that ELF3 acts as a transcriptional repressor of this gene. To test the binding of ELF3 to the *PPD1* promoter *in vivo*, we performed replicated chromatin immunoprecipitation (ChIP) experiments using PS plants carrying the UBI::ELF3-HA transgene combined with the *elf3* mutation and non-transgenic

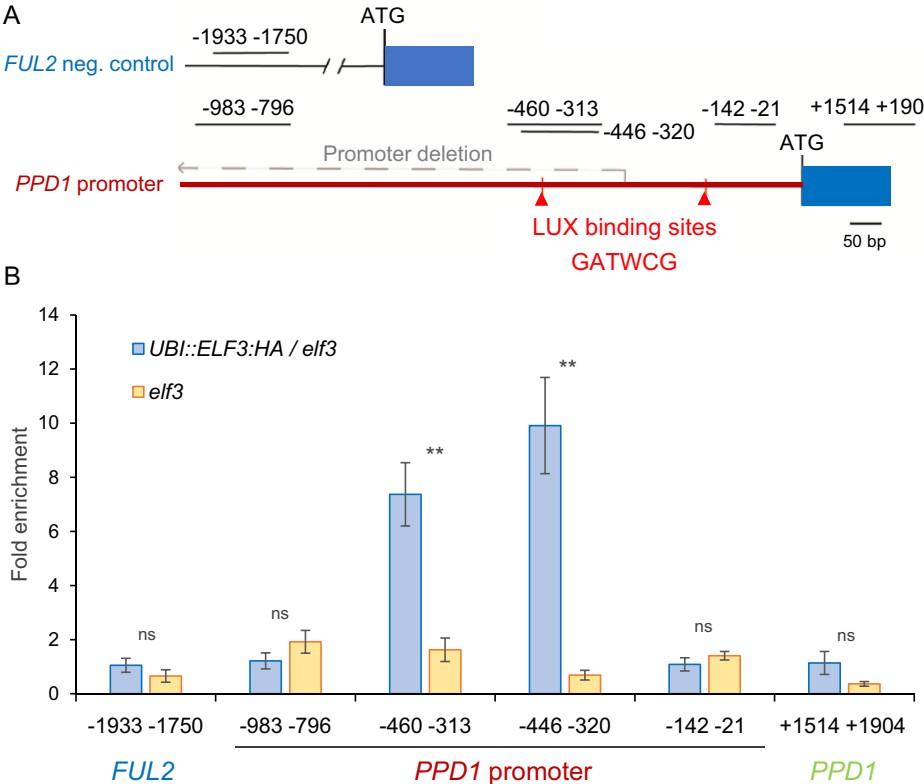

**Fig 8. Chromatin immunoprecipitation (ChIP) analysis of the *PPD1* promoter.** (**A**) Gene diagram of the promoter of *PPD-B1* showing the regions -983 to -796, -460 to -313, -446 to -320, -142 to -21, and a control coding region at +1514 to +1904 analyzed by ChIP, followed by qRT-PCR. The grey dashed arrow demarks the location deleted within the *Ppd-A1a* promoter (PI). The red triangles mark the locations of predicted LUX binding sites (GATWCG). The *PPD1* promoter is indicated by a horizontal red line and a rectangular box represent the first exon (ATG indicates the start codon). (**B**) Fold enrichment of ELF3 at the *PPD1* promoter in the PS-*elf3* mutant and transgenic UBI::ELF3-HA in a mutant PS-*elf3* background. A *FUL* -promoter region between -1933 and -1750 that contained no LUX binding sites was used as a negative control. Bars represent the mean ± s.e.m. from four biological replicate experiments. ** = $P < 0.01$ and ns = not significant. Primer sequences are provided in Table D in S1 Text. Raw data and statistics are in Data G in S1 Data.

*elf3* plants as a control. Samples were collected from the aerial part of plants grown under SD at ZT10 in the dark, when ELF3 levels are elevated (Fig 4).

We observed significant enrichment of ELF3 in two PCR amplified fragments covering *PPD1* promoter regions -460 to -313 and -446 to -320 bp from the start codon (Fig 8 and Table D in S1 Text), which include a LUX binding motif (GATWCG) [36,37]. This suggests that the binding of ELF3 to this promoter region is likely mediated by LUX, which is part of the EC with ELF3 and ELF4. This LUX binding motif (in the minus DNA strand) is deleted in the *Ppd-A1a* allele (Fig 8), likely affecting the binding of the EC.

We did not observe enrichment of ELF3 for a region between -142 to -21 bp (close to another putative LUX binding site), a region in the coding region of *PPD1* between +1514 and +1904 bp, and a region further upstream in the *PPD1* promoter between -983 and -796 bp from the start codon (Fig 8). Lastly, no enrichment of ELF3 was observed at the *FRUITFUL 2* (*FUL2*) promoter (-1933 to -1750), which does not have any putative LUX binding site and was included as a negative control. Taken together, these experiments confirm that ELF3 acts as a direct transcriptional repressor of *PPD1 in planta* (Fig 8), likely as part of the EC.

## Discussion

### *ELF3* regulates wheat heading time and spike development

The role of *ELF3* on the regulation of flowering time has been extensively studied given the importance of this trait in the adaptation of crop species to different latitudes [27,30,47–51]. The early heading time of the *elf3* Kronos mutants in LD and SD and the delayed heading time of the UBI::ELF3-HA transgenic plants indicate that *ELF3* functions as a heading time repressor in wheat. Early flowering of *elf3* mutants has also been reported in barley [27,30] and *Brachypodium* [42], suggesting a conserved function of *ELF3* as a flowering repressor in the temperate grasses. A similar function has been observed in Arabidopsis, where *elf3* mutants flower early under both SD and LD [44,52].

The opposite effect has been observed in rice, where the individual *elf3-1* and *elf3-2* mutants flower later than the wildtype under both SD and LD conditions [53,54] and the double *elf3-1 elf3-2* fails to flower in both photoperiods [39]. In spite of their opposite effects on heading time, the *elf3* mutants show some similarities between wheat and rice. In both species, *elf3* shows increased mRNA levels of the LD flowering repressor *GHD7/VRN2* [53]. In addition, although the *phyC* and *phyB* mutants have opposite effects on heading time in wheat and rice, those are mostly cancelled when combined with the *elf3* mutant in both species [55]. This indicates that the effect of the phytochromes on heading time is mainly mediated by *ELF3* in both wheat and rice [55]. The reverse roles of *ELF3* and the phytochromes in wheat and rice flowering time are likely associated with the reverse role of their downstream target *PPD1/PRR37*, which acts as a LD flowering promoter in wheat and as a LD flowering repressor in rice [56].

The *elf3 ppd1* combined mutant headed earlier than the *ppd1* mutant both in LD (37 d) and SD (>48 d) indicating that *ELF3* can delay wheat heading time independently of *PPD1*. The same result was reported for *Brachypodium* plants grown under 20 h light / 4 h darkness, where *elf3 ppd1* headed 16 d earlier than *ppd1* [57]. However, when the same *Brachypodium* genotypes were grown under SD (8 h light) or LD (16 h light), the *elf3 ppd1* plants headed significantly later (24.4 and 13.7 d) than the *ppd1* plants [57]. These results suggest differences between *Brachypodium* and wheat in the *PPD1*-independent effect of *ELF3*.

In addition to its effects on heading time, the Kronos *elf3* mutants showed a significant reduction in spikelet number per spike (Fig B in S1 Text). Effects of *ELF3* on heading time and spikelet number per spike have been previously reported in diploid wheat *T. monococcum*, where the *ELF3* locus was originally described as *Eps-A^m1* [26,58,59]. Interestingly, the differences in heading time and spikelet number per spike associated with *Eps-A^m1* were both modulated by temperature, an effect that was also reported for the differences in heading time associated with the *EPS-D1* locus in hexaploid wheat [60]. These observations are likely related to the known role of *ELF3* in the temperature entrainment of the circadian clock in Arabidopsis [14,15,36,37,61–63] and barely [64].

### Interactions between *ELF3* and phytochromes *PHYB* and *PHYC*

Previous studies have shown that the wheat photoperiodic response is strongly affected by phytochromes *PHYB* and *PHYC* [11–13] and by *ELF3* [26], which prompted our study of the genetic interactions among these genes and their encoded proteins. The characterization of the different *elf3* and *phyB* combinations revealed that the strong heading time delay observed in the *phyB* mutant almost disappeared in the early heading *elf3 phyB*. The combined mutant also restored the expression of *PPD1*, which was repressed in *phyB*. Similar genetic interactions between *elf3* and *phyC* are reported in the companion study in *Brachypodium* [57], suggesting that these interactions are conserved in the temperate grasses. Taken together, these results

indicate that the critical role of *PHYB* or *PHYC* in the light activation of *PPD1* is mediated by *ELF3* in both wheat and *Brachypodium*.

The genetic interactions between *ELF3* and the phytochromes were also reflected in a physical interaction between the ELF3 protein and both PHYB and PHYC proteins in Y2H assays (Fig 3). Although these interactions still need to be validated *in planta* for wheat, *in planta* interactions between phytochromes and all three members of the EC have been reported before in Arabidopsis, where they have been proposed to be important in the connection between the circadian clock and the light signaling pathways [65,66]. We hypothesize that these physical interactions might be associated with the rapid reduction of ELF3 when the lights were turned on at ZT0, and with the rapid replacement of the ELF3 lower band in the Western blots by a higher and more diffuse ELF3 band when the lights were turned on during the night (Fig 4C). The light modification of the lower ELF3 band was followed by the upregulation of *PPD1* 30 min after the lights were turned on, suggesting that the lower ELF3 band represents the active repressor of *PPD1*. A similar upregulation of *PPD1* was previously reported in wheat after 15 min pulse of light in the middle of a 16 h night [17]. The same study showed that 15 consecutive NBs were sufficient to accelerate wheat heading under SD, but that this acceleration disappeared in the *phyB* and *phyC* mutants [17]. These results suggest that PHYB and PHYC are required for the light modification of ELF3 and the resulting induction of *PPD1* after NBs.

In rice, Western blots using an ELF3 antibody revealed two forms of the ELF3-1 protein, a post-translationally modified higher band detected during the light and dark periods, and a relatively lower band detected only during the night [39]. This lower band was also detected during the day in the rice *phyB* mutant, suggesting that PHYB is necessary for the efficient modification of the lower band by light [39]. The rice ELF3 protein is known to be a direct substrate of the E3 ubiquitin ligase HAF1, which plays a role in the determination of rice heading time under LD [67]. In Arabidopsis, the ubiquitin-ligases CONSTITUTIVELY PHOTOMORPHOGENIC1 (COP1) and XBAT31 can also mediate the ubiquitination and proteasomal degradation of ELF3 [68,69]. These studies suggest that ubiquitination can be responsible for the observed changes in the wheat and rice ELF3 protein when exposed to light during the night, but we cannot completely rule out phosphorylation or other posttranscriptional modifications.

The interactions between the phytochromes and ELF3 in wheat differ from those reported in Arabidopsis, where PHYB contributes to ELF3 accumulation in the light, likely by competing for COP1 and limiting the degradation of ELF3 [70]. The contrasting effects of the phytochromes on ELF3 protein stability may contribute to their different effect in wheat and Arabidopsis flowering time. In the temperate cereals, both the *phyB* and *phyC* mutants flower extremely late under LD [11,13], whereas in Arabidopsis the *phyB* mutants flower earlier under both SD and LD and *phyC* flowers earlier under SD [71]. However, the different photoperiod pathways affected in these species may also contribute to the differences in flowering time caused by the phytochrome mutations. In Arabidopsis, PHYB is involved in the morning degradation of the CO protein, so the *phyB* mutation results in the accumulation of CO and early flowering [2,72]. By contrast, in the temperate cereals PHYB and PHYC operate as flowering promoters in the *PPD1*-dependent photoperiod pathway, and the mutants are extremely late [11,13,73,74].

## Interactions between *ELF3* and *PPD1*

The *ppd1* mutant delays heading time even in an *elf3* background, indicating that it operates downstream of *ELF3*. This is also evident in the changes of *PPD1* transcription profiles observed in the *elf3* mutants (Fig 6A, 6B, 6D and 6E). The upregulation of *Ppd-A1b* and *Ppd-*

*B1b* (intact promoter) at ZT0 in the absence of *ELF3* (Fig 6D, PS) is mimicked in the wild type by the *Ppd-A1a* allele carrying the promoter deletion in the presence of *ELF3* (Fig 6A, PI). A similar upregulation at dawn has been observed in the *Ppd-D1a* allele [6] and in a different *Ppd-A1a* allele [8], all carrying overlapping deletions in the *PPD1* promoter (Fig A in S1 Text). Our ChIP-PCR experiments support the hypothesis that this is the result of the direct transcriptional regulation of *PPD1* by ELF3 (Fig 8). A similar interaction has been reported in *Brachypodium* [75], rice [39] and maize [51] suggesting that is conserved in the grass species.

The ELF3 repression of *PPD1* transcription is likely the result of the binding of the EC to the LUX binding site present in the region deleted in the promoter of the *Ppd-A1a* allele. This is supported by the dawn upregulation of *PPD1* in *lux* mutants in barley [31], diploid wheat *T. monococcum* [33,76], and hexaploid wheat with combined mutations in all three *LUX* homoeologs [32]. LUX is a member of the EC that has a specific DNA-binding domain targeting the GATWCG motif and related promoter elements in Arabidopsis [37,38,77,78]. Therefore, the similar effects of *elf3* and *lux* on the transcriptional regulation of *PPD1* suggests that the EC plays an important role in the repression of *PPD1* during the night and at dawn in the temperate grasses. In Arabidopsis, it has been shown that the EC represses the expression of *PRR7* (a close homoeolog of wheat *PPD1*), *PRR9*, *GIGANTEA*, and *LUX* itself [37,38,77,78].

The *PRR3* –*PRR7* paralogs in Arabidopsis and the *PRR37* (*PPD1*)–*PRR73* paralogs in the grass lineage belong to the same clade, suggesting a common history [79,80]. However, since the duplication that originated *PRR3* and *PRR7* in Arabidopsis is independent of the duplication that originated *PPD1*—*PRR73* in the grasses, the potential sub-functionalization of these two pairs of paralogs is also independent [79]. In Arabidopsis, *PRR7* is an important component of the circadian clock with no major effects on the photoperiodic regulation of flowering, whereas natural mutations in *PPD1* selected during wheat [6,8,18,81] and barley [7,82] domestication have a strong effect on the photoperiodic regulation of heading time but limited effects on the transcription profiles of the central oscillator [46,79]. However, transformation of the Arabidopsis *prr7-11* mutant with the barley photoperiod sensitive (*Ppd-H1*) and insensitive (*ppd-H1*) alleles driven by the Arabidopsis *PRR7* promoter complements circadian leaf movements in Arabidopsis [83], suggesting a conserved clock function. We hypothesize that selection during wheat and barley domestication may have favored *PPD1* mutations with limited effects on the clock that minimize negative pleiotropic effects. Taken together, these results suggest that grasses may have repurposed an ancient interaction between clock genes (EC and *PRR7*) to develop a photoperiod pathway that can operate independently of *CO*.

## Interactions between *ELF3* and *CO1 / CO2*

In a previous study, we showed that combined loss-of-function mutants *co1 co2* headed 3 d earlier than the wildtype under LD and 13.5 d under SD, suggesting that *CO1* and *CO2* function as weak heading time repressors in the presence of functional *PPD1* alleles [5]. However, under LD the *ppd1* mutant headed more than 60 d earlier than the *ppd1 co1* plants, which failed to head before the experiment was terminated at 180 d [5]. A similar effect was reported in barley, where a strong QTL for heading time under LD was found at the *CO1* locus, but only among accessions carrying the *ppd-H1* photoperiod insensitive allele [84]. These results demonstrate that in wheat and barley *CO1* can accelerate heading time under LD in the absence of *PPD1*, and may explain the residual photoperiodic response observed in the *elf3 ppd1* mutant, which headed earlier under LD than under SD (Fig 5).

A comparison of the *CO1* expression results in wheat and *Brachypodium* revealed differences between these two species. In wheat, *CO1* is upregulated in *phyB* (Fig C in S1 Text) and *ppd1* (Fig 7K), whereas in *Brachypodium CO1* is downregulated in *phyC* and *ppd1* [57]. These

differences may explain the earlier heading of the *co1* mutant relative to the wildtype in wheat [5], and the delayed heading of *CO1* RNA interference knock-down plants in *Brachypodium* under LD [85]. Differences in *CO1* regulation may also contribute to the contrasting effect of *ELF3* in the *ppd1* background under 16h LD, where *elf3 ppd1* headed 36 d earlier than *ppd1* in wheat but 2.5 d later in *Brachypodium*. In summary, even though *Brachypodium* and wheat have similar photoperiodic genes, changes in the interactions among these widely interconnected genes can result in different flowering outcomes.

## A working model for the wheat photoperiod pathway

Based on the interactions among flowering genes described in this and previous studies, we proposed a working model of the wheat photoperiodic pathways that is presented in Fig 9. In this model, we place ELF3 between the phytochromes and *PPD1*, as a critical component of

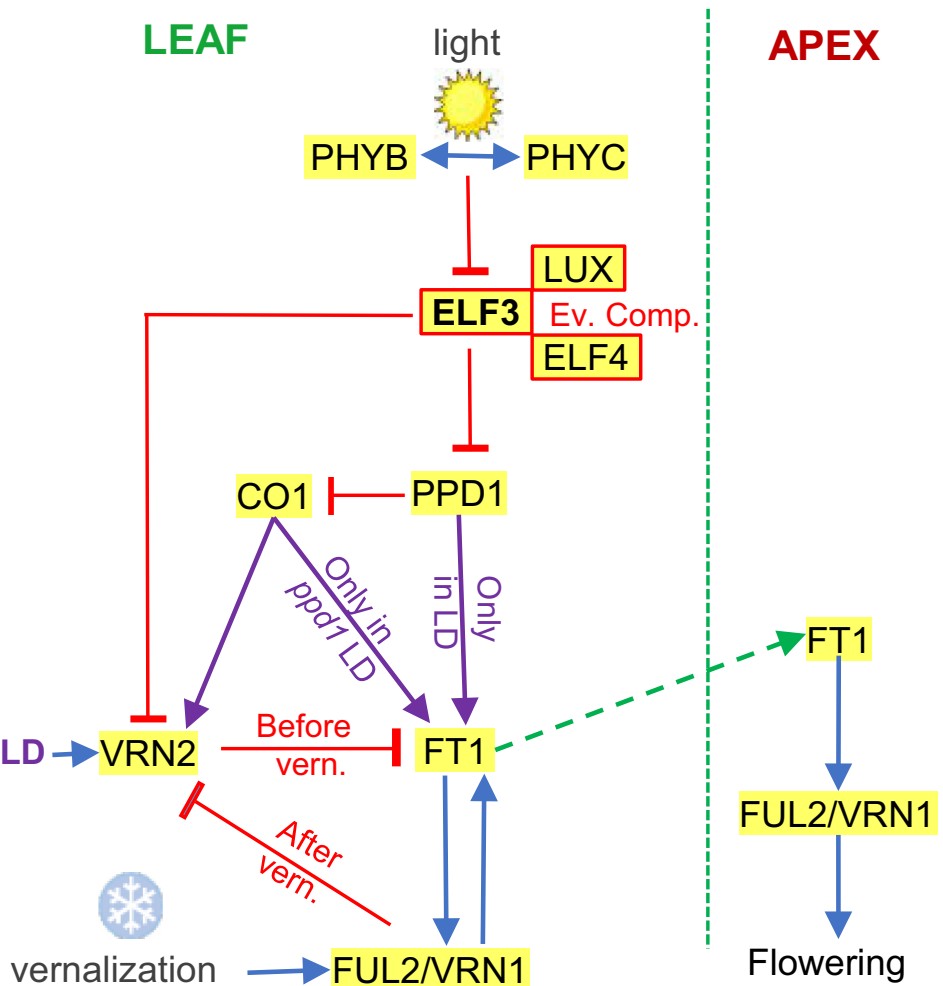

**Fig 9. Working model for the regulation of heading time in wheat.** The model integrates photoperiod and vernalization signals into the regulation of *FT1* expression in wheat leaves. FT1 is then transported to the shoot apical meristem (dotted green arrow) where it induces the transition from the vegetative to the reproductive phase. Blue arrows indicate promotion of gene expression or activity and red lines ending in a crossed-bar indicate repression. ELF3, LUX and ELF4 proteins form the evening complex, which binds to the *PPD1* promoter and inhibits its transcription. The complex interactions between *VRN2* and *CO1*, *CO2* and *PPD1* [5] are not included in the figure for clarity.

the transcriptional activation of *PPD1* by light. We hypothesize that a PHYB-PHYC heterodimer is required to repress the activity of ELF3 since mutations in both phytochromes result in very late flowering and reduced *PPD1* expression [11].

In this model, ELF3 acts as a transcriptional repressor of *PPD1* by direct binding of the EC to a LUX-binding site in the *PPD1* promoter, with the promoter deletions in *Ppd-A1a* and *Ppd-D1a* limiting the ability of the EC to repress their expression. This model explains the upregulation of *PPD1* in the *elf3* (Fig 6) and *lux* mutants at dawn [31–33,76], and its repression in the UBI::ELF3-HA transgenic plants (Fig E in S1 Text).

According to this model, *PPD1* is required for the upregulation of *FT1* under LD, but this induction requires at least two weeks of NBs or LD to induce the expression of *FT1* and accelerate heading time [17]. The NB experiments also suggest that the connection between *PPD1* and *FT1* may be gated by circadian clock-regulated genes. Although the maximum acceleration of heading time occurs when the NBs are applied in the middle of the night, the maximum induction of *PPD1* occurs when the NBs are applied at the end of the night [17]. The proposed gating mechanism of the PPD1 activity is also supported by the induction of *FT1* transcription under LD but not under SD in spite of the induction of *PPD1* transcription during the light phase both under SD and LD conditions [5,17].

PPD1 interferes with CO1 function, which induces *FT1* and accelerates heading time under LD only in a *ppd1* mutant background [5]. CO1 also acts as a transcriptional promoter of *VRN2* [5], which may explain the upregulation of *VRN2* transcript levels observed in the *ppd1* mutant (Fig 7H). The interactions between *VRN2* and both *CO1* and PPD1 are not shown in the model for simplicity. *VRN2* is also negatively regulated by ELF3 (Fig 7G and [28]), an interaction that has been also reported in rice. In this species, OsELF3 promotes flowering under LD mainly by direct repression of *GHD7*, the rice ortholog of *VRN2* [39,54,86]. In wheat, *VRN2* acts as a LD flowering repressor of *FT1* [24], so the low transcript levels of *FT1* in *elf3 ppd1* (Fig 7C) can be caused by a combination of high levels of *VRN2* and absence of *PPD1*.

A limitation of the model presented in Fig 9 is the absence of *GIGANTEA* (*GI*), which has not yet been functionally characterized in wheat. Loss-of-function mutants in *GI* have a reduced photoperiodic response in both rice [87] and Arabidopsis [88]. In Arabidopsis, ELF3 is known to interact with COP1 *in vivo* to promote GI protein degradation [68]. In addition, ChIP studies have shown that the EC binds to the GI promoter and regulates its transcription [37]. Therefore, interactions between ELF3 and GI can also contribute to the photoperiodic response in the temperate cereals and are an important target for future studies in our laboratory.

Although much remains to be learned on the wheat *PPD1*-mediated photoperiod pathway, this study clearly shows that the ELF3 protein operates downstream of PHYB and PHYC, is regulated by light and acts as a direct repressor of *PPD1*. This information provides additional entry points to engineer heading time in wheat, an important trait for the development of better adapted varieties to a changing environment.

## Materials and methods

### Plant materials

Plant materials used in this study were derived from a sequenced EMS-mutagenized TILLING population of the tetraploid wheat variety Kronos (*Triticum turgidum* ssp. *durum*, 2n = 28, genomes AABB) [89,90]. Kronos is a spring PI wheat that carries a functional *Vrn-A1* allele for spring growth habit (*Vrn-A1c* allele) [23,91], a functional *Vrn-B2* long day repressor locus (*ZCCT-B2a* and *ZCCT-B2b)* [92], and the *Ppd-A1a* allele for reduced photoperiodic response

[8]. Kronos lines carrying loss-of function mutations in the A- and B- genome copies of genes *ELF3* [26], *PHYB* [13], and *PPD1* [17] were described before and are summarized in Fig A in S1 Text.

Mutants for the individual homoeologs are indicated with the capital letter of the genome (e.g. *ppd-A1* and *ppd-B1*) whereas the double mutants without any functional copy of the gene are indicated without any genome letter (e.g. *ppd1*). We intercrossed these different mutants and selected different mutant combinations using marker-assisted selection. All lines had at least two backcrosses to parental line Kronos to reduce the number of background mutations. The *elf3 phyB* combined mutant was also crossed with a Kronos near-isogenic line carrying the photoperiod sensitive *Ppd-A1b* allele [93] to test the mutant combinations in both Kronos-PI (PI) and Kronos-PS (PS) backgrounds.

## Genotyping of mutant lines

Kompetitive Allele Specific PCR (KASP) markers (LGC-Genomics, UK) [94] were developed to detect the EMS-induced nucleotide changes that resulted in premature stop codons in genes *ELF3*, *PHYB*, and *PPD-A1*. The mutants' identification numbers, the type and position of the loss-of-function mutations, and the primers used in the diagnostic KASP assays are listed in Table A in S1 Text. To detect the *PPD-B1* deletion, we used a TaqMan assay described previously [81].

KASP reactions were carried out using 5 μl of genomic DNA diluted to a concentration of 5–50 ng μl$^{-1}$, 5 μl of 2x KASP Master Mix, and 0.14 μl of KASP primer mix. For every 100 μl, the primer mix included 12 μl of the VIC primer (100 μM), 12 μl of the FAM primer (100 μM), 30 μl of the genome-specific common primer (100 μM), and 46 μl of distilled water. A two-step touchdown PCR program was used, with an initial denaturalization step of 94˚C for 15 min, followed by ten cycles of touchdown of 94˚C for 20 s and annealing/extension at 61–55˚C for 1 min (dropping 0.6˚C per cycle), followed by 36 cycles of 94˚C for 20 s and annealing/extension at 55˚C for 1 min. KASP results were analyzed with a FLUOstar Omega F plate reader (BMG Labtech, Ortenberg, Germany) using the software KlusterCaller (LGC Genomics, Teddington, UK).

## Phenotypic characterization of mutant lines

After the two backcrosses to Kronos, BC$_2$F$_3$ plants homozygous for the different allelic combinations were stratified for 3 days at 4˚C in the dark and then planted in PGR15 growth chambers (Conviron, Manitoba, Canada). Lights were set to 350 μmol m$^{-2}$ s$^{-1}$ at canopy level and were kept on for 16 h in the LD experiments and for 8 h in the SD experiments. Temperature was set to 22˚C during light periods and 17˚C during dark periods. Heading time was recorded as the number of calendar days from planting in the soil to full emergence of the spike from the sheath.

## Quantitative reverse transcription-PCR (qRT-PCR) analyses of flowering time genes

The transcriptional profiles of genes regulating wheat flowering time were analyzed during a 24-hour period in PI and PS plants with different mutant combinations. Plants were grown in a Conviron growth chamber under LD conditions and tissue samples from the last fully-expanded leaf were collected simultaneously from five-week-old plants every 4 h, starting at Zeitgeber time 0 (ZT 0 = 6:00 am, lights on). Because of variation in leaf developmental rates of the different mutants, the last fully-expanded leaf at week five was leaf four in *ppd1* and

*ppd1 elf3*; leaf five in wildtype and *phyB*, and leaf seven in *elf3* and *elf3 phyB*. Five plants per genotype were analyzed at each time point. Different plants were sampled for each time point.

Harvested tissue was grounded to a fine powder in liquid nitrogen and RNA was extracted using the Spectrum Plant Total RNA Kit (Sigma-Aldrich, St. Louis, MO, USA) following the manufacturer's recommendations. First-strand cDNAs were synthesized from 500 ng of total RNA using the High Capacity Reverse Transcription kit (Applied Biosystems, Foster City, CA, USA). The qRT-PCR experiments were performed using 2X USB VeriQuest Fast SYBR Green qPCR Master Mix (Affymetrix, Inc., Santa Clara, CA, USA) in a 7500 Fast Real-Time PCR system. Primers used for quantitative qRT-PCR are listed in Table B in S1 Text. *ACTIN* was used as an endogenous control. Transcript levels for all genes are expressed as linearized fold-*ACTIN* levels calculated by the formula $2^{(ACTIN\ CT - TARGET\ CT)}$ ± standard error (SE) of the mean (s.e.m.).

## Yeast-two-hybrid (Y2H) assays

The *ELF3* full-length coding region was first cloned into the pDONR-Zeo vector by BP reaction (Thermo Fisher, http://www.thermofisher) using primers ELF3-GW-F and ELF3-GW-R (Table C in S1 Text). It was then transferred to the pLAW10 vector (with the GAL4 DNA-binding domain) with the LR cloning Gateway strategy (Thermo Fisher, http://www.thermofisher) and used as bait. The cloning of the N-PHYC (1–600 AA) and C-PHYC (601-1139AA) was described previously [11]. The N-PHYB (1-625AA) was cloned into the pGADT7 vector, which contains the GAL4 activation domain, between *Eco*RI and *Xma*I restriction sites using primers N-PHYB-F and N-PHYB-R, and the C-PHYB (626-1166AA) was cloned into pGADT7 between *Eco*RI and *Cla*I sites using primers C-PHYB- F and C-PHYB-R (Table C in S1 Text). The PHYB and PHYC proteins were used as preys. SD medium lacking Leucine and Tryptophan (-L-W) was used to select for yeast transformants containing both bait and prey vectors. Interactions were determined on SD media lacking Leucine, Tryptophan, Histidine, and Adenine (-L-W-H-A).

Autoactivation tests for the ELF3-bait was tested against empty vector pLAW11 as prey. N-PHYC and C-PHYC preys have been previously shown to result in no autoactivation with the empty bait vector pGBKT7 [11] and the N-PHYB and C-PHYB preys were confirmed to show no autoactivation in this study.

## Generation of transgenic wheat plants constitutively expressing *ELF3*

The full-length coding region of ELF3 from *T. monococcum* accession G3116 was fused to a C-terminal 3xHA tag and cloned downstream of the maize *UBIQUITIN* promoter in the binary vector pLC41, which has the kanamycin resistance for selection in *E. coli* and *Agrobacterium*, and hygromycin resistance for selection in planta. *Agrobacterium* strain EHA105 was used to infect Kronos immature embryos using the Japan Tobacco transformation technology licensed to UC Davis. The following primers were used to confirm the presence of the transgene: Ubi-F2 CAGAGATGCTTTTTGTTCGC and ELF3-genotyping-R3 AAAGCCTCCCAGATGTAGCA.

## Immunoblot analysis

Tissue from newly fully-expanded leaves was harvested from UBI::ELF3-HA transgenic plants. Protein extraction was performed as described previously [39]. Whole leaf tissue was ground to powder in liquid nitrogen. For each timepoint, 50 mg of powder was transferred into a 1.5 mL microcentrifuge tube with 150 ul of 2 x Laemmli buffer (Bio-Rad, Hercules, CA, USA). The sample was immediately vortexed for 1 min followed by incubation at 70˚C for 10 min.

The supernatants were harvested after centrifugation at 20,000 x g for 10 min. The protein extract was subjected to immunoblot analysis with mono-HA antibody. Briefly, proteins were separated on a 7.5% Mini-PROTEAN TGX stain-free precast gel (Bio-Rad, Hercules, CA, USA). Trans-blot TURBO system was used to transfer proteins from the gel to a PVMF membrane. The membrane was blocked with TBST buffer (20 mM Tris, 137 mM sodium chloride, 1% Tween20 (v/v)) with 5% milk powder for 1 h. The blot was incubated in TBST solution containing 1:2000 Anti-HA-Peroxidase (Roche) for 1 h. The blot was washed 3 times, then developed with chemiluminescent detection reagent SuperSignal West Femto (Thermo Fisher, http://www.thermofisher).

## Chromatin immunoprecipitation analysis

Kronos PS plants carrying the *elf3* mutation or a combination of this mutation and the UBI:: ELF3-HA transgene were grown to the fourth-leaf stage under SD (8h light). Eight grams of above-ground seedling tissue (four replicates, 2 g / replicate) were harvested at ZT10 and fixed in formaldehyde. Plants were harvested in the dark prior to fixation to avoid degradation of the ELF3 protein in the light. To aid in fixation penetration, harvested tissue was cut into 2–4 mm pieces and placed into metal tea infusers. Tissue was immediately cross linked under vacuum for 10 minutes in buffer containing 400 mM sucrose, 10 mM Tris (pH 8), 1% formaldehyde, then the vacuum was turned off and the tissue sat for an additional 5 minutes. To quench the cross-linking reaction 0.25 M glycine was used and the vacuum re-applied for an additional 5 minutes. Tissue was rinsed with milli-q water three times. After fixation, tissue was frozen in liquid nitrogen and stored at −80˚C until performing the chromatin immunoprecipitation as described before [95]. Anti-HA magnetic beads from Thermo Fisher (http://www.thermofisher) were used for the immunoprecipitation.

## Supporting information

**S1 Text. Fig A. Schematic representations of *PHYB*, *ELF3*, and *PPD1* mutants.** Mutations introducing premature stop codons are indicated with red triangles and deletions with dotted red lines. Exons are shown as gray rectangles and introns as black lines. (**A**) The *phyB* line combines a premature stop codon on the A-genome homoeolog that truncates the last 641 amino acids of the protein, spanning the entire regulatory module, and a premature stop codon in the B-genome copy that eliminates the distal 140 amino acids, including the histidine kinase domain [13]. (**B**) The *elf3* line carries premature stop codons that eliminate the last 241 (A-genome) and 244 (B-genome) amino acids of the C-terminal region, including the third and fourth conserved blocks of the ELF3 protein [26]. (**C**) The *ppd1* line carries a premature stop codon in *ppd-A1* that eliminates 514 amino acids of the PPD-A1 protein, including the highly conserved CCT domain, and *ppd-B1* is a gamma ray-induced deletion the eliminates the complete gene [5,17]. (**D**) Natural deletions in the promoter of the *Ppd-A1a* [8] and *Ppd-D1a* [6] alleles in tetraploid and hexaploid wheat, respectively. **Fig B. Effect of *elf3* and *elf3 phyB* mutations on spikelet number per spike (SNS).** (**A**) Kronos photoperiod insensitive (PI). (**B**) Kronos photoperiod sensitive (PS). In both experiment plants were grown under LD (16h light). Different letters above the bars indicate significant differences in pair-wise non-parametric Kruskal-Wallis tests ($P < 0.05$). The non-parametric test was used because no transformation was able to restore normality of residuals and homogeneity of variances simultaneously. The reduced SNS in the *elf3* mutant was restored in the *phyB elf3* combined mutant. Raw data is available in Data H in S1 Data. **Fig C. Transcript levels of flowering genes *VRN1*, *VRN2*, *CO1*, and *CO2* in Kronos PI, *phyB* and *elf3 phyB*.** (**A-B**) *VRN1*, (**C-D**) *VRN2*, (**E-F**) *CO1*, and (**G-H**) *CO2*. (**A, C, E,** and **G**) Wildtype *vs. phyB*. (**B, D, F,** and **H**) Wildtype *vs. elf3*

*phyB*. Primers used for qRT-PCR amplify both homoeologs of each gene. The WT data is the same within each row but can be at different scales. Error bars are s.e.m based on 5 biological replications. ns = not significant, * = $P < 0.05$, ** = $P < 0.01$, *** = $P < 0.001$ based on *t*-tests between mutants and wildtype at the different time points. Raw data and statistics are available in Data I in S1 Data. **Fig D. *ELF3* transcript levels in leaves in the presence of *phyB* and *phyC* mutants under different photoperiods.** (**A**) Transcript levels of *Elf3* during the day in PS under SD. (**B**) *Elf-A3* and *Elf-B3* transcripts per million (TPM) in PI WT, *phyB* and *phyC* in leaves collected at ZT4 from 4-w old plants grown under LD and 8-w old plants grown under SD. Data are from previously published RNA-seq [12]. (**C**) Transcript levels during the day in PI and *phyC* under LD (**D**) Transcript levels during the day in PI and *phyB* under LD. Transcript levels were determined by qRT-PCR using *ACTIN* as endogenous control. NS = $P > 0.05$. Raw data and statistics are in Data J in S1 Data **Fig E. Effect of the constitutive expression of *ELF3* under the maize *UBIQUITIN* promoter (UBI::ELF3-HA).** (**A-C**) Transcription profiles in the 7th leaf collected at ZT0, ZT4, ZT12 and ZT20 from five-week-old Kronos PS plants grown under LD. *ACTIN* was used as endogenous control. (**A**) *ELF3*, (**B**) *PPD1* (conserved primers that amplify both *Ppd-A1b* and *Ppd-B1b*) and (**C**) *FT1*. Different letters indicate significant differences in Tukey tests ($P < 0.05$). The colors of the letters match the color of the respective treatment. OE = UBI::ELF3-HA and NT = non-transgenic sister line. (**D**) Complementation of the *elf3* mutant by UBI::ELF3-HA. A factorial ANOVA showed highly significant effects on days to heading (DTH) for both the mutant and transgenic genotypes and for their interaction. Simple effects were tested by contrasts: ns = not significant, ** = $P < 0.01$, *** = $P < 0.001$. Raw data and statistics are available in Data K in S1 Data. **Fig F. ELF3-HA protein size in western blots.** (**A-B**) Samples extracted from leaves of Kronos-PS transgenic plants over-expressing ELF3 and a C-terminal 3xHA tag under the maize *UBIQUITIN* promoter (UBI::ELF3-HA). The ELF3-HA protein was detected by immunoblotting using an anti-HA antibody. Leaf samples were collected from plants grown under short day either 10 minutes before the lights were turned off (ZT8 -10m) or four hours after the lights were turned off (ZT12). (**A**) The expected size of the ELF3-HA protein is 88.5 kDa but the estimated size of the lowest band detected with the HA antibody was ~110 kDa based on both pre-stained and unstained protein markers. No bands were detected in the negative untransformed controls confirming the identity of the ELF3-HA protein. An additional and more diffuse band was detected at ZT8 -10m between 110 and 130 kDa. (**B**) Replicated experiment showing the complete blot. The lower strong band is rubisco (Rub). Top: Chemiluminescence, middle: CBB stained color-matrix, and bottom: merged images obtained by ChemiDoc imaging system (BioRad). (**C**) Wheat protoplasts transformed with UBI::ELF3:HA grown under both light and dark conditions. The unstained molecular marker was used to estimate band size. **Table A**. EMS-induced nucleotide changes that resulted in premature stop codon mutations in *ELF3*, *PHYB*, and *PPD-A1*. The ID of each of the mutant lines, the position of the loss-of-function mutations in the CDS and protein, and the primers used in the diagnostic KASP assays are listed. Capital letters in the primer sequences indicate the VIC and FAM tails. The 3′ allele-specific nucleotides are underlined. **Table B**. Primers used for qRT-PCR analysis. **Table C**. Primers used for cloning ELF3 and PHYB in yeast-two-hybrid assays. **Table D**. Primers used for chromatin immunoprecipitation.
(DOCX)

**S1 Data. Excel File (spreadsheets A to H) including data and statistical analyses supporting figures and supplemental figures. Data A**. Supporting data for Fig 1. **Data B**. Supporting data for Fig 2. **Data C**. Supporting data for Fig 4. **Data D**. Supporting data for Fig 5. **Data E**. Supporting data for Fig 6. **Data F**. Supporting data for Fig 7. **Data G**. Supporting data for Fig

8. **Data H**. Supporting data for Fig B in S1 Text. **Data I**. Supporting data for Fig C in S1 Text. **Data J**. Supporting data for Fig D in S1 Text. **Data K**. Supporting data for Fig E in S1 Text. (XLSX)

## Author Contributions

**Conceptualization:** Maria Alejandra Alvarez, Jorge Dubcovsky.

**Formal analysis:** Maria Alejandra Alvarez, Jorge Dubcovsky.

**Funding acquisition:** Jorge Dubcovsky.

**Investigation:** Maria Alejandra Alvarez, Chengxia Li, Huiqiong Lin, Anna Joe, Daniel P. Woods.

**Project administration:** Jorge Dubcovsky.

**Resources:** Maria Alejandra Alvarez, Chengxia Li, Huiqiong Lin, Anna Joe, Mariana Padilla, Daniel P. Woods.

**Supervision:** Jorge Dubcovsky.

**Writing – original draft:** Maria Alejandra Alvarez.

**Writing – review & editing:** Chengxia Li, Huiqiong Lin, Anna Joe, Daniel P. Woods, Jorge Dubcovsky.

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
