## [Decision Letter · Decision Letter 0]

10 Dec 2022

Dear Dr Dubcovsky,

Thank you very much for submitting your Research Article entitled 'EARLY FLOWERING 3 interactions with PHYTOCHROME B and PHOTOPERIOD1 are critical for the photoperiodic regulation of wheat heading time' to PLOS Genetics.

The manuscript was fully evaluated at the editorial level and by independent peer reviewers. The reviewers appreciated the attention to an important topic but identified some concerns that we ask you address in a revised manuscript.

We therefore ask you to modify the manuscript according to the review recommendations. Your revisions should address the specific points made by each reviewer.

Yours sincerely,

Claudia Köhler

Section Editor

PLOS Genetics

Reviewer's Responses to Questions

**Comments to the Authors:**

Reviewer #1: This is a very interesting and clear manuscript that demonstrates that ELF3 plays a critical role in the wheat photoperiod pathway by regulating the light signal between the phytochromes and PPD1.

Loss-of-function mutations in ELF3 result in the upregulation of PPD1 at and dawn, and in early heading under both long and short days, even in the absence of PHYB. A deletion in the PPD1 promoter including an ELF3 binding region also results in earlier heading under short days, indicating that ELF3 acts as a direct transcriptional repressor of PPD1. The authors generated a large number of mutants and double mutants in different Kronos backgrounds (PI, PS). The paper is clearly written and well structured.

I only have a few minor comments on the manuscript.

1. Fig 1: It was surprising that Ppd1-A and Ppd1-B have a 4h shift in peak expression in the WT. Also how do you explain the very different diurnal pattern of Ppd1-B in KPI and KPS?

2. Fig 4. How do you explain that in the night break experiment (C) the second Elf3 band starts appearing only 10min after lights on, whereas in LD (Fig A) after lights on no elf3 can be seen (the gel at ZT0+10min looks different from the rest of the gel background?). Would this not suggest that not only the light but also the time of the day (circadian, or Phy night conversion) would matter for elf3 protein abundance and form? Is your idea that the lower band elf3 in the night is the active one?

3. Since you have the elf3-HA would it not be possible to really show binding of elf3 to phys in vivo?

4. Fig. 5 Please, explain in the M&M part what exactly is ppd1-null is and how this line was generated.

5. On page 17 you write: The notable exception is the downregulation of VRN2, CO1, and CO2 at dawn in elf3 phyB (S3D, F, and H) relative to elf3 (Fig7G, J and M), suggesting the PHYB is required for the upregulation of these genes in the elf3 mutant background at dawn.

6. In S3D VRN is upregulated in elf3 phyb, actually in all mutants VRN2 is upregulated, also in the phyb mutant and this is somewhat puzzling. It might also be indirect effects through feedback loops between flowering time genes (Ppd1, VRN1, CO, VRN2, FT1) as is actually indicated in your flowering model.

7. As the authors provide a model for a barley flowering time pathway with ELF3 in a central position, but also discuss the possible role of LUX, it might be nice to add LUX to the model in Fig. 9

Reviewer #2: The authors provide a concise genetic analysis of photoperiodic flowering in the temperate cereal wheat. Through the phenotypic and molecular analysis of single and double mutants they show that ELF3 acts downstream of PHYB in the induction of the flowering activator PPD1, with ELF3 being a negative regulator of PPD1 expression. They further show that PHYB does not affect ELF3 gene expression but rather interacts with the PHYB photoreceptor (and PHYC) (in yeast two-hybrid), suggesting that PHYB regulates ELF3 activity, possibly by light-mediated post-translational modification of the ELF3 protein. They further show by ChIP that ELF3 directly binds the promoter of PPD1.

This is a very thoughtful study. The data support the conclusions drawn. The manuscript is well-written. This manuscript stands out because it not only genetically analyzes ELF3 function but also addresses a possible mechanism of ELF3 regulation by studying the ELF3 protein.

I only have a few minor comments:

Fig. 4A. In this Western blot, the ELF3-HA protein runs at about 130 kDa. However, ELF3 is predicted to be only approx. 75 kDa in size. That is quite a discrepancy. I ask the authors to show a control blot with a protein extract of a non-transgenic plant as a control in order to confirm that the detected band(s) are indeed ELF3-HA. In the supplement, please also show a complete blot including the region of the gel where 75 kDa proteins are expected.

I wonder if the authors have confirmed the PHYB-ELF3 interaction using additional methods? E.g. split luciferase, BiFC. This would strengthen the manuscript. However, since this interaction is described for Arabidopsis, I do not require these experiments.

Fig. 4: The authors speculate that the upper ELF3 band might reflect PIF3 phosphorylation or ELF3 ubiquitination. A treatment of the extract with phosphatase could easily test for phosphorylation and would strengthen the manuscript.

**Have all data underlying the figures and results presented in the manuscript been provided?**

Reviewer #1: Yes

Reviewer #2: Yes

PLOS authors have the option to publish the peer review history of their article (what does this mean?). If published, this will include your full peer review and any attached files.

Reviewer #1: No

Reviewer #2: No

---

## [Decision Letter · Decision Letter 1]

4 Feb 2023

Dear Dr Dubcovsky,

We are pleased to inform you that your manuscript entitled "EARLY FLOWERING 3 interactions with PHYTOCHROME B and PHOTOPERIOD1 are critical for the photoperiodic regulation of wheat heading time" has been editorially accepted for publication in PLOS Genetics. Congratulations!

Yours sincerely,

Claudia Köhler

Section Editor

PLOS Genetics

Claudia Köhler

Section Editor

PLOS Genetics

Comments from the reviewers (if applicable):

Reviewer's Responses to Questions

**Comments to the Authors:**

Reviewer #1: The authors have responded to all my earlier questions and remarks. I have no further concerns or questions.

Reviewer #2: My comments were well-addressed by the authors.

**Have all data underlying the figures and results presented in the manuscript been provided?**

Reviewer #1: Yes

Reviewer #2: Yes

PLOS authors have the option to publish the peer review history of their article (what does this mean?). If published, this will include your full peer review and any attached files.

Reviewer #1: No

Reviewer #2: No

**Data Deposition**

http://datadryad.org/submit?journalID=pgenetics&manu=PGENETICS-D-22-01175R1

**Press Queries**

---

## [Editor Report · Acceptance letter]

25 Apr 2023

PGENETICS-D-22-01175R1 

EARLY FLOWERING 3 interactions with PHYTOCHROME B and PHOTOPERIOD1 are critical for the photoperiodic regulation of wheat heading time 

Dear Dr Dubcovsky, 

We are pleased to inform you that your manuscript entitled "EARLY FLOWERING 3 interactions with PHYTOCHROME B and PHOTOPERIOD1 are critical for the photoperiodic regulation of wheat heading time" has been formally accepted for publication in PLOS Genetics! Your manuscript is now with our production department and you will be notified of the publication date in due course.

With kind regards,

Zsofia Freund

PLOS Genetics

On behalf of:
